# Effects of Endochondral and Intramembranous Ossification Pathways on Bone Tissue Formation and Vascularization in Human Tissue-Engineered Grafts

**DOI:** 10.3390/cells11193070

**Published:** 2022-09-29

**Authors:** Jonathan C. Bernhard, Darja Marolt Presen, Ming Li, Xavier Monforte, James Ferguson, Gabriele Leinfellner, Patrick Heimel, Susanna L. Betti, Sharon Shu, Andreas H. Teuschl-Woller, Stefan Tangl, Heinz Redl, Gordana Vunjak-Novakovic

**Affiliations:** 1Department of Biomedical Engineering, Columbia University, New York, NY 10032, USA; 2Ludwig Boltzmann Institute for Traumatology, The Research Center in Cooperation with AUVA, 1200 Vienna, Austria; 3Austrian Cluster for Tissue Regeneration, 1200 Vienna, Austria; 4Department of Life Science Engineering, University of Applied Sciences Technikum Wien, 1200 Vienna, Austria; 5School of Dentistry, Medical University of Vienna, 1090 Vienna, Austria; 6Department of Medicine, Columbia University, New York, NY 10032, USA; 7College of Dental Medicine, Columbia University, New York, NY 10032, USA

**Keywords:** endochondral, intramembranous, ossification, bone tissue engineering, mesenchymal stromal cells

## Abstract

Bone grafts can be engineered by differentiating human mesenchymal stromal cells (MSCs) via the endochondral and intramembranous ossification pathways. We evaluated the effects of each pathway on the properties of engineered bone grafts and their capacity to drive bone regeneration. Bone-marrow-derived MSCs were differentiated on silk scaffolds into either hypertrophic chondrocytes (hyper) or osteoblasts (osteo) over 5 weeks of in vitro cultivation, and were implanted subcutaneously for 12 weeks. The pathways’ constructs were evaluated over time with respect to gene expression, composition, histomorphology, microstructure, vascularization and biomechanics. Hypertrophic chondrocytes expressed higher levels of osteogenic genes and deposited significantly more bone mineral and proteins than the osteoblasts. Before implantation, the mineral in the hyper group was less mature than that in the osteo group. Following 12 weeks of implantation, the hyper group had increased mineral density but a similar overall mineral composition compared with the osteo group. The hyper group also displayed significantly more blood vessel infiltration than the osteo group. Both groups contained M2 macrophages, indicating bone regeneration. These data suggest that, similar to the body’s repair processes, endochondral pathway might be more advantageous when regenerating large defects, whereas intramembranous ossification could be utilized to guide the tissue formation pattern with a scaffold architecture.

## 1. Significance Statement

Bone tissue engineering protocols have utilized MSCs differentiated into both osteoblasts and hypertrophic chondrocytes. Overall, engineered bone grafts have performed inconsistently in animal studies of bone regeneration, due, at least in part, to the unknown contributions of the differentiated MSCs. In the current study, we conducted a detailed in vitro and in vivo comparison of bone grafts engineered from human bone marrow MSCs induced into the two ossification pathways. We identified important differences in the gene expression, composition, microstructure and vascularization of the two ossification pathways utilized in bone graft development, which can be used to shape the regenerative engineering strategies for bone. These findings aid our understanding of the relative advantages of replicating the endochondral vs. the intramembranous ossification pathways for bone regeneration.

## 2. Introduction

Mesenchymal stromal cells (MSCs) play an integral role in bone formation, maintenance and repair through their ability to differentiate in response to environmental cues, as well as their paracrine activity [1,2]. In situations where bone regeneration requires an external intervention, such as in the treatment of critical size defects, MSCs can dictate the progress of repair through their interactions with the healing milieu [3]. Tissue engineers have studied MSCs in a number of regenerative strategies to create viable and transplantable bone grafts [4]. One direction of this research has focused on recapitulating the signals present during native bone formation in order to engineer bone grafts with enhanced functional properties [5].

A majority of the earlier studies aimed to recapitulate the intramembranous ossification pathway, with MSCs differentiating directly into osteoblasts depositing a mineralized bone matrix [6]. Cultivation of the MSC-seeded scaffolds in bioreactors provided mechanical signals to the cells, and enhanced the development of mineralized bone-like tissue [7,8,9]. However, despite the robust osteogenic potential of MSCs observed in tissue-engineered grafts cultured in vitro, the in vivo results have been variable. In some studies, differentiated osteoblasts promoted the integration of the engineered graft and stimulated angiogenesis, with limited graft resorption [10,11,12]. In other studies, mechanical failure of the grafts, poor vascularization and only minimal bone formation were reported [13], indicating the limitations of this bone tissue engineering approach. More recently, induction of the endochondral ossification pathway has been investigated, where MSCs were directed into the chondrogenic lineage and then subjected to hypertrophic maturation [14,15,16].

Hypertrophic chondrocytes are known to express bone formation genes and deposit significant amounts of mineral [15,16]. When implanted, hypertrophic chondrocytes enhanced bone matrix turnover and regeneration [16,17]. Hypertrophic chondrocytes also promoted angiogenesis, while the induction of the endochondral ossification pathway was shown to support the formation of bone organs with a functional hematopoietic stem cell niche [17]. These characteristics suggested that the bone grafts grown using hypertrophic chondrocytes might present advantages over osteoblast grafts for successful bone regeneration. Alternatively, the induction of the two differentiation pathways might be tailored to meet the requirements of specific conditions of bone repair.

In recent studies, differentiation into hypertrophic chondrocytes and osteoblasts was evaluated for the ability to promote the healing of critical size calvarial and femoral defects [18,19,20]. In calvarial defects in rats, hypertrophic chondrocytes derived from rat bone marrow MSCs on collagen–hyaluronic acid scaffolds supported greater and more homogenous bone repair than the osteoblasts differentiated on collagen–hydroxyapatite scaffolds [20].

However, the vascularization of the engineered constructs was comparable between the two groups. In a recent study, human bone marrow MSCs that were primed briefly into the endochondral and intramembranous pathways on polycaprolactone scaffolds exhibited comparable new bone formation in murine calvarial defects, whereas blood vessel infiltration was higher in the endochondral group [19]. In femoral defects of rats, human MSCs derived from adipose tissue and induced into hypertrophic chondrocytes on decellularized trabecular bone scaffolds enhanced integration, remodeling and defect bridging, compared with the MSCs induced into osteoblasts and acellular scaffold controls [18].

Due to the discrepancies among these studies, we were interested in directly comparing the in vitro and in vivo behavior of bone marrow MSCs differentiated into the two ossification pathways to better understand their capacity for bone formation in vitro and regeneration in vivo. We hypothesized that the cells’ properties, deposition of the matrix and the resulting potential of the engineered constructs for bone formation and regeneration would strongly depend on the cells’ differentiation regime. To test this hypothesis, we induced bone marrow MSCs into osteoblasts and hypertrophic chondrocytes using biologically inert silk scaffolds, which exhibited high potential for bone tissue engineering in our previous studies and demonstrated a muted inflammatory tissue reaction upon implantation [7,21,22]. After we had analyzed the gene expression profiles, cytokine secretion profiles and bone matrix deposition patterns, the engineered constructs were investigated for their regenerative potential in a subcutaneous implantation model for 12 weeks.

## 3. Materials and Methods

Sigma Aldrich (St. Louis, MO, USA) was the source for all purchased materials. The purchased materials were of analytical grade unless otherwise specified.

### 3.1. Silk Scaffold Fabrication

Silk solutions for scaffold generation were prepared at a concentration of 16 wt% bombyx mori silk fibroin in hexafluoroisopropanol (HFIP), following published methods [23]. To generate a porous scaffold, the HFIP silk solution was poured over NaCl salt 400–600 µm in diameter and allowed to solidify. Silk NaCl scaffolds were then submerged in methanol for 1–2 days to induce ß-sheet formation. Finally, the NaCl was washed out of the scaffolds with diH_2_O for 2 days. Cylinders 2 mm in height and 4 mm in diameter were cut from the material and the scaffolds were disinfected with 70% ethanol.

### 3.2. Cell Expansion, Seeding and Differentiation

Human bone marrow MSCs (Lonza, Basel, Switzerland) were acquired, characterized, and expanded as in previous studies [21]. The expansion medium contained a high-glucose medium with L-glutamine, 1 ng/mL basic fibroblast growth factor, 10% fetal bovine serum and 1% penicillin/streptomycin. Sterile cylindrical scaffolds were submerged in an expansion medium for a single day before seeding. MSCs at Passage 5 were trypsinized and resuspended in an expansion medium at a concentration of 3 × 10^7^ cells/mL. Scaffolds were seeded with 750,000 cells per scaffold using the drip method [7]. MSC-seeded scaffolds were kept in the expansion medium for an additional day, and then the medium was changed according to the appropriate cultivation regime.

Constructs mimicking intramembranous ossification (termed osteo constructs) were generated by facilitating MSC differentiation over five continuous weeks in an osteogenic medium. The osteogenic medium was composed of low-glucose DMEM (ThermoFisher, Waltham, MA, USA), supplemented with 10% fetal bovine serum, a 10 mM HEPES buffer, 1% penicillin/streptomycin (P/S), 50 µg/mL ascorbic acid, 100 nM dexamethasone and 5 mM β-glycerophosphate. Generated in parallel to the constructs mimicking intramembranous ossification, constructs mimicking endochondral ossification (termed hyper constructs) were created by first differentiating the MSCs in chondrogenic medium for two weeks to generate chondrocytes. The chondrogenic medium consisted of high-glucose DMEM (ThermoFisher) supplemented with 50 µg/mL ascorbic acid, 100 nM dexamethasone, 50 µg/mL proline, 1% ITS+, 100 µg/mL sodium pyruvate, 1% P/S, 10 ng/mL TGF-β3 and 10 ng/mL BMP6. After two weeks of chondrogenic differentiation, the MSC-derived chondrocytes were matured over three weeks into hypertrophic chondrocytes. The hypertrophic maturation medium was composed of high-glucose DMEM supplemented with 50 µg/mL ascorbic acid, 1 nM dexamethasone, 50 µg/mL proline, 1% ITS+, 100 µg/mL sodium pyruvate, 1% P/S, 5 mM β-glycerophosphate and 50 ng/mL L-thyroxine.

### 3.3. Real-Time PCR

After five weeks of in vitro cultivation, the RNA was extracted by utilizing TRIzol (ThermoFisher) according to the manufacturer’s instructions. NanoDrop spectrophotometric quantitation (ThermoFisher) was utilized to determine the total RNA content in each scaffold. DNase treatment was utilized to remove contaminating DNA, and then a high-capacity cDNA reverse transcription kit (ThermoFisher) was used to transcribe the RNA into cDNA, according to the manufacturer’s instructions. Quantitative RT-PCR was performed with the transcribed cDNA using the Fast Sybr Green mix (ThermoFisher). All samples (*n* = 4) from each experimental group were evaluated in duplicate. The ΔC_t_, defined by the C_t_ of GAPDH subtracted from the C_t_ of the gene of interest, was utilized to normalize the quantitations. Table 1 presents the forward and reverse primers for each gene of interest.

### 3.4. Biochemical Assays

Constructs harvested at the culmination of five weeks were first weighed and then digested with papain (40 units/mg) in a digestion buffer at 60 °C overnight. The digestion buffer consisted of 10 mM cysteine HCl, 0.1 M sodium acetate and 50 mM EDTA at pH 6.0. Quantitation of the DNA in each scaffold digest was analyzed using a Quant-iT PicoGreen assay kit (ThermoFisher) following the manufacturer’s instructions. The manufacturer-supplied lambda DNA standard was used in the quantitation, with each sample’s amount of DNA normalized to the sample’s wet weight.

Dimethylmethylene blue assay was utilized to determine the amount of sulfated glycosaminoglycan (GAG) deposited within each scaffold, consistent with prior publications [18]. Chondroitin-6-sulfate served as the control.

A modified protocol [24] was leveraged to determine the collagen content within each scaffold through quantitation of hydroxyproline. The papain digest from each sample was incubated overnight at 110 °C in 6 M HCl and then reacted with chloramine T and Elrich’s reagent. The amount of hydroxyproline within the samples was assessed spectrophotometrically at 540 nm and determined utilizing a standard hydroxyproline curve.

To assess the calcium content, the harvested constructs were snap-frozen in liquid nitrogen and stored at −20 °C. A 5% trichloroacetic acid solution was utilized to extract the calcium from the constructs, and the sample was quantified using the Calcium (CPC) Liquicolor kit (Stanbio Laboratory, Boerne, TX, USA) following the manufacturer’s instructions.

### 3.5. Fourier Transform Infrared Spectroscopy (FTIR)

A Nicolet iS5 (ThermoFisher) with a diamond window containing an iD5 ATR accessory was utilized to record the FTIR spectra. Samples were embedded in paraffin, and sections (6 µm thick, 15 for each sample) were sliced for analysis. The FTIR spectra of the samples were measured using scans with a resolution of 400–4000 cm^−1^. The specific regions of interest (900–1200 cm^−1^ and 1590–1720 cm^−1^) were isolated, baselined and normalized using Omnic software (ThermoFisher). A genetic algorithm that fit up to 12 Gaussian distributions was created with Matlab software (Mathworks, Natick, MA, USA) to de-convolute the normalized data. The peak intensity of the sub-peaks of interest was recorded for each measurement. The presence of mineral within the matrix was measured by analyzing both the phosphate band in the curve in the 900–1200 nm range and the Amide I band in the curve present in the 1600–1700 nm range. The maturity of the mineral, determined by the extent of the mineral’s crystallinity, was assessed by calculating the ratio of the sub-band area at 1030 nm to the sub-band area at 1020 nm.

### 3.6. Mechanical Testing

Previously published methods [7] were utilized to determine the equilibrium modulus and the Young’s modulus for each construct group (*n* = 6). To summarize, an initial compressive load of 0.2 N was applied to the sample of interest, followed by a stress relaxation step. The compression during the step occurred at a ramp velocity of 1% per second until a total strain of 10% was reached, with the compressive load then held at 10% strain for 1800 s. The Young’s and equilibrium moduli were calculated from the measured forces.

### 3.7. Subcutaneous Implantation

After five weeks of in vitro cultivation, the osteo and hyper constructs (both experimental groups) were inserted into subcutaneous pouches in nude mice, following previously approved animal experimentation procedures. The constructs were harvested after 3, 6 and 12 weeks after implantation and analyzed as described below.

### 3.8. Micro-Computed Tomography (µCT)

Upon completion of in vitro differentiation and prior to implantation, a vivaCT 40 system (Scanco Medical, Bruttisellen, Switzerland) was utilized to scan the constructs (*n* = 4) from each experimental group. The scans followed a protocol modified from previously published methods [25]. The vivaCT’s settings in the modified protocol were: current, 0.109 mA; voltage, 55 kV; slice thickness, 21 µm; isotropic resolution, 21 µm. The data thresholds for three-dimensional reconstructions and quantitation were set at 220 mg HA/cm^3^. Scanco Medical software was utilized to calculate the bone volume, the bone mineral density (BMD) and the bone surface to volume ratio (BS/BV).

Constructs harvested at 3 weeks (*n* = 4), 6 weeks (*n* = 6) and 12 weeks (*n* = 4) were scanned on a µCT 50 system (Scanco Medical, Bruttisellen, Switzerland) utilizing the following scanner settings: current, 0.200 mA; voltage, 70 kV; slice thickness, 10 µm; isotropic resolution, 10 µm. A global thresholding technique set at 282.9 mg HA/cm^3^ was utilized for three-dimensional reconstructions and quantitation. Software provided by Scanco Medical was used to calculate the bone volume, BMD and BS/BV.

### 3.9. Histology and Immunohistochemistry

Pre-implantation constructs for histology and immunohistochemistry were preserved through incubation for 24 h in 4% formaldehyde, rinsed with PBS, dehydrated in a series of solutions with ascending ethanol concentrations, embedded in paraffin and sectioned at 6 µm per slice. Hematoxylin and eosin staining was used to examine the histomorphology of the constructs. Von Kossa staining of the constructs was utilized to visualize the presence of phosphate in the deposited mineral. Alkaline phosphatase (ALP), osteopontin (OPN) and bone sialoprotein (BSP) were assessed through immunohistochemistry. Separately, the hyper constructs were stained with Alcian Blue for the presence of glycosaminoglycans (GAG), and collagen type 10 (COLX) was evaluated by utilizing immunohistochemistry for hypertrophic chondrocyte matrix deposition to confirm mimicking of the endochondral ossification process. Antigen retrieval was carried out by submerging the samples in a citrate buffer in a container which was placed into a boiling water bath for 20 min. Incubation in 0.3% hydrogen peroxide for 30 min was utilized to block the samples, which were then washed and incubated overnight at 4 °C with the primary antibodies against BSP (1/500 dilution) (EMD Millipore, Bilerica, MA, USA), OPN (1/500 dilution) (EMD Millipore), ALP (1/250 dilution) and COLX (1/2000 dilution) (AbCam, San Francisco, CA, USA). The Vectastain Elite Universal staining kit (Vector Laboratories, Burlingame, CA, USA) was utilized to detect primary antibodies. Counterstaining was completed with Hematoxylin QS (Vector Laboratories).

After harvest and scanning with the μCT, subcutaneously implanted constructs were cut into halves. One half of each construct was used for a decalcified histological evaluation. These samples were preserved in 4% formaldehyde for 24 h, rinsed with PBS and decalcified using a formic-acid-based solution (Immunocal Decalcifier, StatLab, McKinney, TX, USA). After the samples had been decalcified, they were rinsed with PBS, dehydrated in ethanol solutions, embedded in paraffin and sectioned at 6 µm. Antigen retrieval was conducted on the samples, and the Vectastain ABC Kit for Rabbit IgG (Vector Laboratories) was used for secondary antibody staining. Primary antibodies for M1 (iNOS, 1/100 dilution) (AbCam) and M2 (Arg1, 1/100 dilution) (AbCam) macrophages were incubated overnight at 4 °C, as well as the primary antibody for CD31 (1/50 dilution) (AbCam). Samples were counterstained with Hematoxylin QS (Vector Laboratories) after visualization of the primary antibody. The number of blood vessels, the distance of the blood vessels from the scaffold edge and the areas of the constructs and the blood vessels were determined for the constructs harvested after 6 weeks (*n* = 6) through semi-quantitation analysis by two independent blinded researchers using NIH ImageJ (Version 1.47, http://rsb.info.nih.gov/ij/, (Accessed on 2 February 2016)).

The remaining half of the subcutaneous implants were processed for hard tissue histology. Samples incubated in 4% formaldehyde for 24 h were used to preserve the constructs, followed by rinsing in PBS and dehydration in a series of solutions with ascending ethanol concentrations before the samples were embedded in light-curing resin (Technovit 7200 VLC; Kulzer & Co., Wehrheim, Germany). Undecalcified and thinly ground sections were prepared along the longitudinal axis of the constructs according to Donath [26] and stained with Levai–Laczko dye as previously described [27]. Histological specimens were visualized and scanned with the Olympus dotSlide 2.4 digital virtual microscopy system (Olympus, Tokyo, Japan) at a resolution of 0.32 µm per pixel.

### 3.10. Statistical Analyses

All data are presented as the mean ± standard deviation. Significance between the pre-implantation and post-implantation constructs was determined using Student’s *t*-test, α = 0.05, with the significance determined by *p* < 0.05. Analysis was conducted with Prism Software (GraphPad, La Jolla, CA, USA). Post-implantation µCT quantitation was analyzed through one-way analysis of variance (ANOVA), followed by Tukey’s post-hoc test, α = 0.05, with the significance determined by *p* < 0.05 for each construct type. Differences in the distribution of the vessels’ distance from the edge were assessed between the two ossification pathways by ANOVA, α = 0.05, with the significance determined by *p* < 0.05.

## 4. Results

Human bone marrow MSCs were seeded on porous silk scaffolds, differentiated by inducing the endochondral and intramembranous ossification pathways for 5 weeks and implanted subcutaneously into nude mice for up to 12 weeks (Figure 1). All the MSCs utilized in the constructs and experiments were from the same initial pool of MSCs, with ossification pathway differentiation occurring in parallel.

At the end of cultivation, the engineered tissue constructs from both groups had comparable DNA and collagen contents, but differed in their calcium content, glycosaminoglycan (GAG) content and matrix deposition patterns (Appendix A). The expression profiles of key bone-related marker genes differed significantly between the hyper and osteo constructs (Figure 2A). The expression of RUNX2, a master regulator of MSCs’ osteogenic differentiation, was significantly higher in the hyper constructs compared with the osteo constructs. The increased expression of RUNX2 correlated with the increased expression of genes for bone matrix proteins, including Collagen Type I (COL1A1); the mineralization-related proteins bone sialoprotein (IBSP), alkaline phosphatase (ALPL), and osteonectin (SPARC); and the bone remodeling proteins osteocalcin (BGLAP) and osteopontin (SPP1). Notably, the localization patterns of bone matrix proteins also differed between the two groups (Figure 2B). In the hyper constructs, BSP, OPN and ALP were deposited in the extracellular space surrounding the enlarged chondrocytes’ lacunae throughout the constructs, whereas in the osteo constructs, these proteins were co-localized with the osteoblasts’ nuclei in close proximity to the silk scaffolds’ surfaces. In the hyper constructs, the deposition of glycosaminoglycans and Collagen Type X, and the increased expression of hypertrophy-related genes confirmed the differentiation into the endochondral pathway (Appendix A).

Differences in the mineralization patterns were observed between the two groups of constructs, as shown by von Kossa staining (Figure 2B). In the hyper constructs, mineral was present in small nodules that fused together and were localized within the scaffolds’ pores. In the osteo constructs, larger mineral deposits were located on the surfaces of the scaffold surfaces that were undergoing mineralization. An analysis of the mineralized matrix by FTIR (Figure 2C) showed that the mineral to matrix ratio, an indicator of the degree of mineral present within the matrix, was not significantly different between the hyper and the osteo constructs (1.58 ± 0.7 to 1.27 ± 0.6, respectively). In contrast, the mineral crystallinity ratio, which is indicative of the mineral’s maturity, was significantly higher in the osteo constructs compared with the hyper constructs (1.1 ± 0.6 and 0.5 ± 0.3, respectively).

Finally, differences in Young’s elastic and equilibrium moduli were measured through compression tests for the two groups (Appendix A). While the mechanical properties of the osteo constructs did not increase with cultivation, the hyper chondrocyte constructs acquired significantly higher Young’s and equilibrium moduli over the course of 5 weeks of cultivation.

As cellular signaling mediates the complex process of bone regeneration, the cytokines released in the conditioned media of both construct groups were analyzed (Figure 3A). The hyper constructs were found to release a greater amount of bone morphogenetic protein-2 (BMP-2), BMP-6 and BMP-7. In contrast, the concentrations of some pro-inflammatory cytokines (TNF-α, IFN-γ) were higher in the conditioned medium of the osteo constructs. Cytokines related to osteoclastogenesis and bone resorption (IL-6, IL-8, MCP-1) were also released at higher concentrations from the osteo constructs, whereas the release of BMP-4, IL-1ß, RANK, RANKL and M-CSF was comparable between the groups.

The hyper and osteo constructs were implanted subcutaneously in nude mice to evaluate the influence of each differentiation pathway on ectopic bone formation. To quantify the amount of bone deposition and visualize the development of mineralized tissue, samples were scanned before implantation and after 3, 6 and 12 weeks using µCT. Before implantation, the three-dimensional representation of mineral deposition (Figure 3B and Appendix A) corresponded to the von Kossa staining (Figure 2B). Namely, the hypertrophic chondrocytes deposited nodules of mineral clustered closely together in bulbous depositions in the interior of the construct. The osteo constructs had sparser, spindle-like mineral depositions spread around the constructs’ exterior (Appendix A).

Upon implantation, these constructs underwent remodeling that differed between the two groups. The hyper constructs did not deposit significantly more mineral (i.e., increase bone volume) during the 12 weeks in vivo, but instead displayed a significant increase in BMD (Figure 3B). Visually, the mineral appeared to be remodeled from primarily bulbous nodular depositions to a more refined structure with smooth surfaces. In contrast, the osteoblast constructs showed a significant increase in the amount of mineral deposited by 12 weeks. Interestingly, mineral resorption occurred during the first 6 weeks of implantation, whereas over the final 6 weeks in vivo, large amounts of mineral were deposited, with the deposited mineral resembling the architecture of the original scaffold. In contrast to the hyper constructs, the osteo constructs did not show increase in BMD. Furthermore, the hyper constructs maintained a steady BS/BV, whereas in the osteo constructs, the BS/BV value peaked at 3 and 6 weeks, consistent with a dispersed mineral deposition pattern.

Bone tissue formation and remodeling were evaluated in constructs explanted after 3, 6 and 12 weeks using the undecalcified bone histology and Levai–Laczko staining (Figure 4). After 3 weeks, calcified nodular deposits of extracellular matrix were widespread within the pore spaces of the hyper constructs, as indicated by the purple/blue tint of the matrix staining (Figure 4, hyper (top), white asterisk). This nodular deposition pattern was similar to that in the pre-implantation constructs (von Kossa staining, Figure 2B). By 6 weeks, continued mineralization and compacting of the calcified nodular deposits had occurred (Figure 4, hyper (middle), black asterisk), surrounded by purple and less compacted nodular deposits. By 12 weeks, the hyper constructs contained areas of mature bone undergoing remodeling, as indicated by the presence of cement lines (Figure 4, hyper (bottom), white arrow) and embedded osteocytes. Some instances of nodular deposits exhibiting a pink/purple tint were present within the matrix. In contrast, the osteo constructs had only minimal amounts of pink-stained mineral deposition within the marginal regions of the scaffold after 3 weeks (Figure 4, osteo (top), black hash sign). By 6 weeks, the mineral deposition at the margins was more widespread (Figure 4, osteo (middle), black hash sign), and select instances of scaffold mineralization were present (Figure 4, osteo (middle), black arrow). By 12 weeks, the osteo constructs exhibited substantial mineralization of the scaffolds (Figure 4, osteo (bottom), black arrow), corresponding to the findings of the µCT evaluation (Figure 3).

To develop a better understanding of the regenerative environments of both construct groups after implantation, constructs harvested after 3 weeks were stained for the presence of M1 and M2 macrophages. M1 macrophages, staining positively for iNOS, could not be detected in either group of constructs (Appendix A), even though the osteo constructs had an elevated concentration of inflammatory and macrophage-related cytokines (Figure 3A). In contrast, M2 macrophages staining positively for Arg 1 were present in both types of constructs after 3 weeks of implantation (Figure 5). In the hyper constructs, M2 macrophages were localized in the pore spaces surrounding the calcified cartilage-like nodules, whereas in the osteo constructs, they were located near the scaffolds’ surfaces.

Angiogenesis of tissue-engineered constructs is essential for successful integration and bone defect regeneration. Vascular endothelial growth factor A (VEGF-A), which stimulates and directs angiogenesis, was found at a significantly higher concentration in the conditioned medium of the osteo constructs compared with that of the hyper constructs prior to implantation (Figure 6A). In contrast, immunohistochemical staining revealed that after 6 weeks in vivo, CD31+ blood vessels were significantly more abundant in the hyper constructs compared with the osteo constructs (Figure 6B,C and Appendix A). To determine the degree of vessel ingrowth in the constructs, the distances from the vessels’ center to the nearest construct edge were evaluated. The vessels were found to have grown significantly more deeply into the hyper constructs compared with the osteo constructs (Appendix A), with the maximum distance of a vessel from the scaffold’s edge being recorded in the hyper constructs (Appendix A). Furthermore, the vessels were found to be relatively uniformly distributed in the interior of the hyper constructs (Figure 6D). In contrast, the osteo constructs contained the majority of vessels within the first 200 µm from the constructs’ surfaces.

## 5. Discussion

Bones form through two main pathways, endochondral and intramembranous ossification. Endochondral ossification occurs in situations that require substantial tissue deposition or the establishment of mechanical stability, and is characterized by the initial deposition of the cartilage matrix, followed by hypertrophic chondrocyte-mediated bone deposition [28]. In contrast, intramembranous ossification occurs within the osteoblast-generated matrix and is associated with osteoblast-mediated remodeling [29,30]. The ability to mimic the native pathways of bone development in tissue-engineered constructs has been shown previously [7,8,10,16,17,31]. In endochondral ossification, the initial differentiation of bone marrow MSCs into chondrocytes produces a glycosaminoglycans-rich cartilaginous matrix, which serves as a basis for the deposition of hypertrophic chondrocyte-specific Collagen Type X and matrix mineralization. In intramembranous ossification, the combination of dexamethasone and ascorbic acid prompts MSCs to differentiate into osteoblasts, which directly deposit a bone-like mineralized matrix.

The in vitro differentiation of the bone marrow MSCs into the two pathways within silk fibroin scaffolds resulted in significantly different construct properties due to the differentiation pathways, including differences in the gene expression levels of osteogenic markers, the deposition patterns of bone matrix proteins, mineralization and growth factor/cytokine secretion. Regarding the gene expression levels, previous in vitro studies have demonstrated the dynamic changes that occur across various timepoints during differentiation. After 5 weeks of differentiation, the hyper constructs had significantly higher expression levels of RUNX2, the master regulator of osteogenic differentiation [32,33] and of genes for bone matrix proteins, including proteins involved in matrix mineralization and remodeling.

The increased gene expression of bone matrix proteins was reflected in the protein level, with the hyper constructs exhibiting dense protein deposits within the bone scaffolds’ pores, which exceeded the deposits observed in the osteo constructs on the scaffolds’ surfaces. These differences in the gene expression and matrix deposition patterns between the two groups could potentially be attributed to the different characteristics of the differentiation pathways. Intramembranous ossification occurs on an established matrix, with the attachment and subsequent polarization of the osteoblasts dictating the direction and location of bone matrix deposition [34,35]. The necessity of this attachment-mediated polarization may explain the unique location of bone proteins and mineral along the scaffold, and the overall limited quantity of mineral deposition within the osteo constructs. In contrast, endochondral ossification is characterized by the formation of new bone during tissue regeneration, and is initiated through the condensation of MSCs and the subsequent chondrocyte differentiation and cartilage matrix production [36,37,38].

Large pores in the silk scaffolds allowed for the recapitulation of endochondral ossification, as demonstrated by the dense matrix containing GAG and COLX within the pore spaces. The bone matrix proteins and mineral deposits within these large pores did not present any structural patterns, except that the matrix surrounded enlarged cell lacunae, which is characteristic of hypertrophic chondrocytes and is similar to native endochondral ossification deposition patterns [39].

The µCT analysis of the cultured constructs revealed that the mineralized matrix was organized into small spindles in the osteo constructs and into aggregated clumps throughout the hyper constructs. These characteristics are consistent with previous studies [17,18,20,21]. The hyper constructs also demonstrated improved mechanical properties, likely due to the specific morphology and higher amounts of GAG and the COLX, which has been shown to improve mechanical properties [40,41]. In contrast, the osteo constructs did not show an increase in their mechanical properties, which were in similar range to the values previously reported for silk-based osteo constructs [21].

The observed differences in the matrix morphology were accompanied by differences in mineral deposition, with the hyper constructs acquiring significantly more calcium/DNA than the osteo constructs during the 5-week culture period. The increased mineral deposition correlated with lower crystallinity, i.e., mineral maturity, in the hyper constructs. The presence of GAG within the matrix deposition environment is known to disrupt mineral formation, leading to smaller crystallinity values [42,43]. Interestingly, the FTIR values measured in the osteo constructs matched those reported for human cancellous bone [44], suggesting a similarity in the mineral composition between the native bone and the engineered osteo constructs.

Upon implantation, the hyper constructs underwent extensive remodeling, and the immature mineralized matrix deposits were transformed into cortical-like bone containing cement lines and embedded cells, similar to a previous report [20]. The mineralized cartilage matrix serves as a key stage in endochondral ossification and provides a necessary localization point for remodeling and new deposition [45], with hypertrophic chondrocytes potentially undergoing transdifferentiation to aid in their turnover [46,47]. The hyper constructs in our study exhibited a greater release of several BMPs which play a direct role in bone induction and formation, and which presumably aided in this remodeling [48,49]. The survival of hypertrophic chondrocytes in hypoxic environments likely contributed to the strong regeneration shortly after subcutaneous implantation, and is consistent with the bone regeneration results reported in large defect studies [18].The osteo constructs released more pro-inflammatory and bone resorption cytokines, potentially enhancing the osteoclasts’ activity and contributing to the reduction in bone volume at the 3-week timepoint [50]. This reduction was followed by an increase in bone volume up to 12 weeks in vivo, similar to some studies with MSC-based osteo constructs [51,52] and in contrast to other studies [20]. An evaluation of the resorption through hard bone histology demonstrated instances of scaffold resorption caused by mononucleated cells indented at the scaffolds’ surfaces throughout the osteo constructs. However, the lack of multinucleated cells, which are indicators of rapid degeneration, suggest slow silk scaffold resorption and matrix turnover. The bone mineralization in the osteo constructs followed the contour of the structural pattern of the silk scaffold, suggesting potentially greater control of the shape of bone reconstruction through intramembranous ossification processes in tissue-engineered constructs. This behavior was consistent with prior studies [7].

In an attempt to better understand the remodeling of the hyper constructs in vivo, we also explored the presence of macrophages. It has been shown that macrophages, and M2 macrophages in particular, are essential for endochondral ossification, as they play unique roles in bone regeneration [53,54]. We found that the hyper constructs contained large numbers of M2 macrophages co-localized with the calcified cartilage, suggesting a role of hypertrophic chondrocytes in macrophage differentiation and homing. The osteo constructs also contained M2 macrophages, although in distinctly lower amounts than the hyper constructs. A possible reason could be the lower release of BMP-7 by the osteo constructs, as this factor plays a role in the differentiation of M2 macrophages [55]. Interestingly, the presence of M1 macrophages was minimal in both construct groups at 3 weeks, suggesting that the matrix environment was predominantly pro-regenerative at this timepoint. As the constructs were implanted subcutaneously into nude mice, a more comprehensive characterization of the immune-specific response of the mimicked ossification pathways was not investigated. However, future development of hyper and osteo constructs should try to understand the different interactions between these tissue-engineered constructs and the host. Silk fibroin scaffolds are recommended for this evaluation, as previous reports have noted the bioinert properties and muted immune response to silk fibroin scaffolds [22].

Finally, it has been shown that vascularization is critical for bone formation and regeneration in vivo [56], and the differences in the bone formation of the two types of constructs could possibly be impacted by the constructs’ vascularization once transplanted. The hyper constructs that exhibited the widespread presence of blood vessels underwent extensive bone remodeling, whereas the osteo constructs exhibited delayed bone formation that was associated with limited vascularization. The increased vascularization of the hyper constructs is in agreement with a recent report demonstrating that hypertrophic chondrocyte enhanced the constructs’ vascularization in calvarial defects [19]. The differences in VEGF release between the current study and a previous report that resulted in comparable vascularization of both groups [20] may be attributed to the different levels of hypertrophy, as it has been reported that VEGF release decreases with the progression of hypertrophy [16].

One limitation of this study involved the limited characterization of the undifferentiated MSCs within the silk scaffolds as a comparator. While the focus of this study was on the deposition and behavior caused by the mimicked ossification pathways, future studies should attempt to understand the matrix deposition and gene expression of the undifferentiated MSCs compared with the differentiated pathways. A more comprehensive knowledge of undifferentiated vs. differentiated MSCs’ behavior will permit the tailored development of more complex bone tissue engineered constructs.

## 6. Conclusions

The implementation of two differentiation pathways—endochondral and intramembranous ossification—resulted in significantly different deposition patterns of the mineralized bone matrix in tissue constructs engineered from bone marrow MSCs on porous silk scaffolds. The hyper constructs quickly deposited immature mineral and instigated extensive remodeling in vivo, accompanied by enhanced vascularization and the elevated presence of M2 macrophages. The osteo constructs deposited mature mineral along the scaffolds’ surfaces and instigated vascularization at the constructs’ periphery, accompanied by initial bone resorption and followed by bone formation. These responses were reflective of the differences in the signaling components for the two construct types. As the goal of our study was to compare the performance and properties of the two different differentiation regimes to aid future translation, we propose that the differentiation of hypertrophic chondrocytes could be utilized when extensive bone deposition, mechanical strength and vascularization are needed. Direct differentiation into osteoblasts could be utilized in more complex settings, in which slow and mature tissue remodeling can take place, and where greater control of the new tissue’s formation patterns and bone shape can be guided through the scaffolds’ architecture.

## Figures and Tables

**Figure 1 cells-11-03070-f001:**
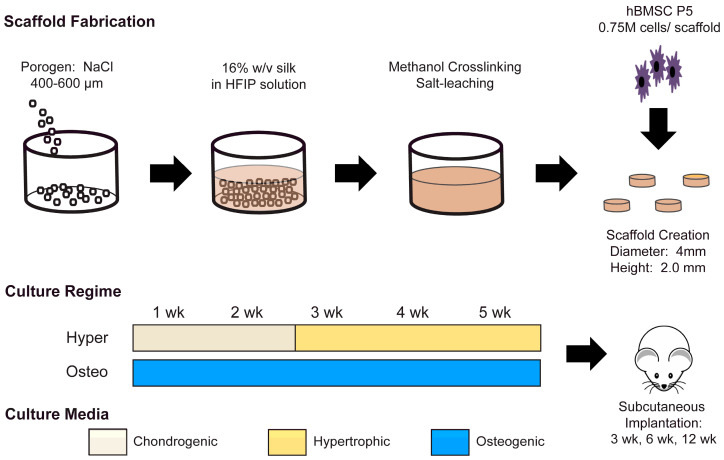
Experimental design. Porous cylindrical scaffolds (4 mm in diameter × 2 mm high) were made from a silk fibroin solution using the HFIP-based salt-leaching technique, and seeded with human bone marrow MSCs from a consistent pool. The MSCs were differentiated in parallel into two ossification pathways. Hypertrophic chondrocyte constructs (Hyper) were derived from MSCs by cultivation in a chondrogenic medium for 2 weeks and then matured to hypertrophic chondrocytes over 3 weeks in a hypertrophic medium to mimic endochondral ossification. Osteoblast constructs (Osteo) were cultured for 5 weeks in an osteogenic medium to generate osteoblasts and mimic intramembranous ossification. After in vitro cultivation, scaffolds from each ossification pathway were implanted subcutaneously in nude mice and harvested for analysis at 3 weeks, 6 weeks, and 12 weeks after implantation.

**Figure 2 cells-11-03070-f002:**
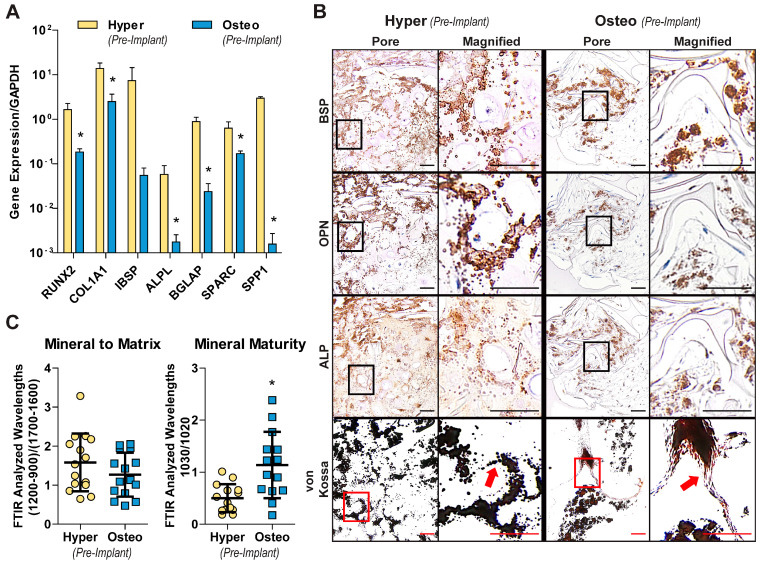
Differentiated cells’ behavior and matrix deposition. (**A**) Expression of the key genes involved in bone development in hypertrophic chondrocyte and osteoblast constructs. The expression levels were normalized to GAPDH (*n* = 4). (**B**) Histology and immunohistochemistry at the scaffold pore level and cellular level (magnified images). Representative images are shown for von Kossa, BSP, OPN and ALP staining to demonstrate the differences in the deposition patterns. Red arrows point to the differences in the mineral’s appearance, with the hyper constructs containing globular mineral deposits and the osteo constructs depositing mineral along the scaffolds’ surfaces. Scale bars: 50 µm. (**C**) Mineral composition within the newly formed bone matrix, as determined by FTIR analysis of the mineral–matrix ratio and the maturity of the mineral (*n* = 16). Data (**A**,**C**) are shown as the average ± SD. * significant differences between the groups (*p* < 0.05).

**Figure 3 cells-11-03070-f003:**
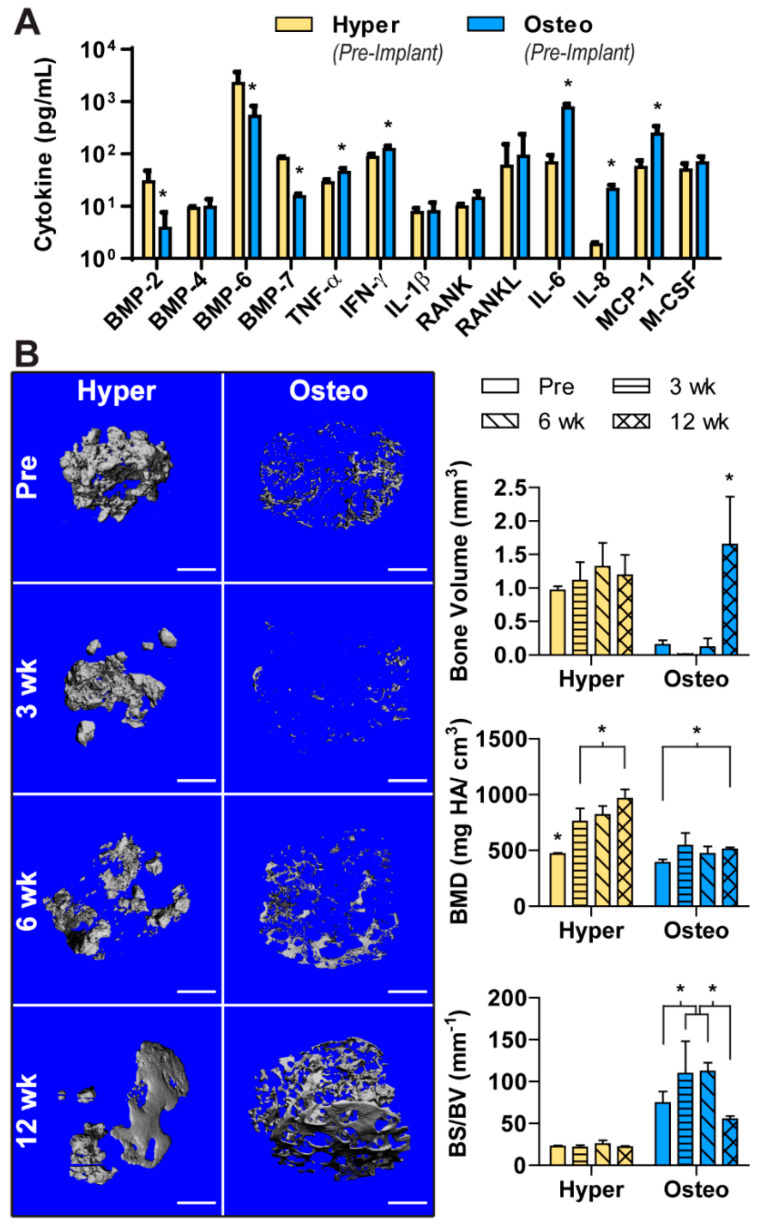
Cytokine release from constructs and in vivo bone formation. (**A**) Cytokines released into the culture medium. The hyper constructs released higher levels of BMP-2, BMP-6 and BMP-7 and lower levels of inflammatory (TNF-α, IFN-γ) and degradation (IL-6, IL-8, MCP-1) cytokines compared with the osteo constructs. The release of BMP-4, IL-1ß, RANK, RANKl and M-CSF was comparable between the groups. (**B**) µCT of the constructs before implantation and following 3, 6 and 12 weeks of implantation. Scale bars: 1 mm. The total bone volume, BMD and BS/BV were determined for each group and timepoint. Data are shown as the average ± SD (*n* = 4). * Significant differences between the groups (*p* < 0.05).

**Figure 4 cells-11-03070-f004:**
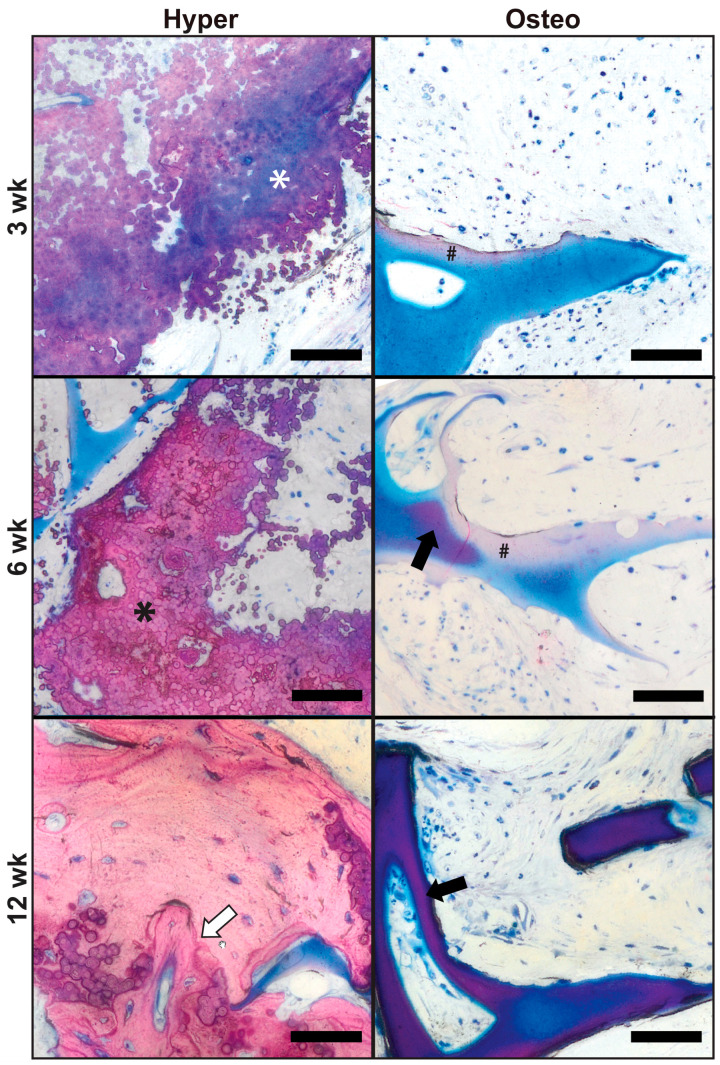
In vivo bone regeneration. The undecalcified bone histology of harvested constructs was conducted following 3, 6 and 12 weeks of implantation. The samples were stained with Levai–Laczko stain to evaluate the presence of mineral and bone regeneration. In the hyper constructs, calcified nodular deposits exhibiting a blue/purple tint (white asterisk) were present at 3 weeks. Continued mineralization and compaction of the extracellular matrix nodules resulted in a change in the tint to a pink color (black asterisk) in the hyper samples at 6 weeks. Mature bone remodeling was indicated by the presence of a cement line (white arrow) in the hyper samples at 12 weeks. In the osteo samples, mineral deposition was initiated in the margins of the silk scaffold, as seen in the samples at 3 and 6 weeks as light pink staining (black hash sign). Dense scaffold mineralization in the osteo samples was present in limited amounts at 6 weeks (dark purple stain, indicated by the black arrow) and was more widespread at 12 weeks, consistent with the µCT evaluation. Scale bars: 50 µm.

**Figure 5 cells-11-03070-f005:**
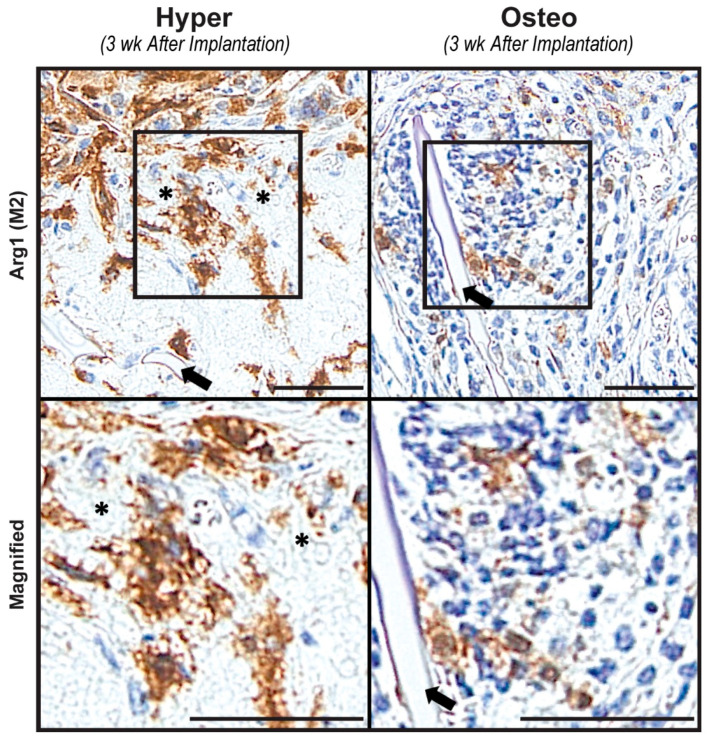
Presence of macrophages. Arg1 immunohistochemistry staining was conducted to evaluate the presence of M2 macrophages in the constructs after 3 weeks of implantation. Representative images demonstrate numerous M2 macrophages, as indicated by the brown staining localized around the matrix resembling calcified cartilage (the white, round matrix marked by black asterisks) in the hyper constructs. In the osteo constructs, fewer M2 macrophages were found to be localized in proximity to the scaffold (the silk scaffold is indicated by black arrows in both constructs). Scale bars: 50 µm.

**Figure 6 cells-11-03070-f006:**
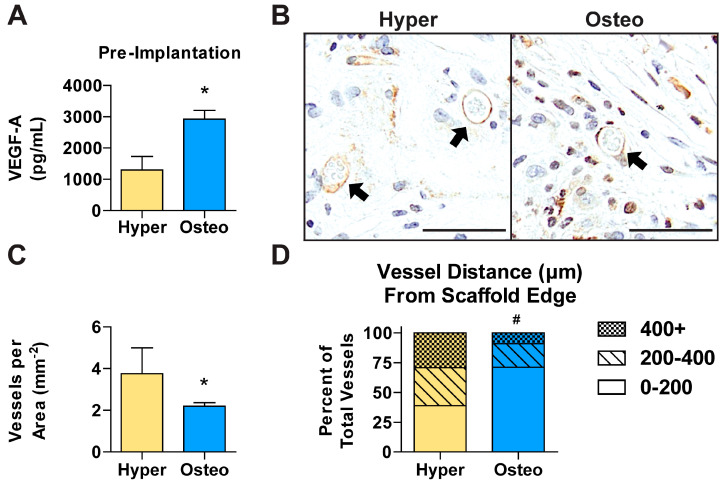
Vascularization of the constructs. (**A**) The concentration of VEGF-A was determined in the culture media of both the hyper and osteo constructs. (**B**) CD31 immunohistochemistry at 6 weeks demonstrated the presence of vessels (black arrows) within the constructs. (**C**) Semi-quantitative analysis of CD31+ vessels per construct area. (**D**) Histogram showing the distance of the vessels from the nearest scaffold edge, grouped into three distance ranges: 0–200 µm, 200–400 µm and above 400 µm. Scale bars: 50 µm. All data (*n* = 4) are shown as the average ± SD. * Significant differences between the groups (*p* < 0.05), ^#^ Significant differences in the distribution of the vessels’ distances from the nearest scaffold edge (*p* < 0.05).

**Table 1 cells-11-03070-t001:** Primers utilized in the RT-PCR evaluation.

Gene	Forward	Reverse
*GAPDH*	AAGGTGAAGGTCGGAGTCAAC	GGGGTCATTGATGGCAACAATA
*RUNX2*	CCGTCTTCACAAATCCTCCCC	CCCGAGGTCCATCTACTGTAAC
*COL1A1*	GATCTGCGTCTGCGACAAC	GGCAGTTCTTGGTCTCGTCA
*IBSP*	GAACCTCGTGGGGACAATTAC	CATCATAGCCATCGTAGCCTTG
*ALPL*	GGGACTGGTACTCAGACAACG	GTAGGCGATGTCCTTACAGCC
*BGLAP*	GGCGCTACCTGTATCAATGG	GTGGTCAGCCAACTCGTCA
*SPARC*	CCCAACCACGGCAATTTCCTA	CGTCTCGAAAGCGGTTCC
*SPP1*	GTTTCGCAGACCTGACATCCA	GCTTTCCATGTGTGAGGTGAT
*COL2A1*	AGACTTGCGTCTACCCCAATC	GCAGGCGTAGGAAGGTCATC
*COL10A1*	CATAAAAGGCCCACTACCCAAC	ACCTTGCTCTCCTCTTACTGC
*MMP13*	CCAGACTTCACGATGGCATTG	GGCATCTCCTCCATAATTTGGC
*IHH*	AACTCGCTGGCTATCTCGGT	GCCCTCATAATGCAGGGACT

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
