# Peer review of "Effects of Endochondral and Intramembranous Ossification Pathways on Bone Tissue Formation and Vascularization in Human Tissue-Engineered Grafts"

_cells, 2022, doi:10.3390/cells11193070_

Round 1
Reviewer 1 Report
Bone grafts are performed inconsistently in animal studies of bone regeneration due at least in part to the unknown contributions of the differentiated MSCs. In this paper, the authors engineer bone grafts using biologically inert silk scaffolds. After analyzing gene expression profiles, cytokine secretion profiles, and bone matrix deposition patterns, the engineered constructs were investigated for their regenerative potential in a subcutaneous implantation model.
Fig 2A. It is better to provide a baseline control for the MSCs without any hyper or osteo construct.
Fig 3A. Comparing the level of cytokines between hyper and osteo construct is important, but it is essential to provide the baseline control for the MSCs without hyper or osteo treatment. This will help to sort out the applications of silk scaffold in intramembranous ossification or endochondral ossification, or both.
Fig 6. Again, adding the control without a culture medium of hyper and osteo constructs is better to prove the advantage of silk scaffolds.
It will be beneficial if the authors have data to indicate the immune reaction after grafting silk scaffold in vivo. Otherwise, I suggest adding some sentences to cover the potential effect of immune response in the discussion.
In summary, the findings are significant and have great potential for translational application.
Reviewer 2 Report
In this study, bone regeneration capacity was compared between the two types of bone grafts induced by different ossification processes. Human bone marrow stromal cells were differentiated on silk scaffold into the hypertrophic chondrocytes or osteoblasts in vitro. Both constructs were implanted subcutaneously and evaluated their gene expression, microstructures, histology, vascularization and so on. As the results, hypertrophic chondrocytes induced more bone and blood vessels than osteoblasts. Therefore, they suggest endochondral ossification pathway would be more advantageous to repair the larger defects.
The result is interesting and the experiment was done well, but some revisions would help us understand the study better.
1. Histology
1) I recommend the authors to show H-E staining sections together with the photos of immunohistochemistry and indicate the scaffold surface. Because the counterstain does not seem to be effective enough to show the cell/tissue morphology in the photos shown in Figure 2B.
2) In Figure 2B, Osteo-von Kossa, not only deposits along the scaffolding but also spherical deposits are found. Does this mean that some cartilage was also formed? 
3) In Figure 5, the 'mineralized cartilage-like matrix within the hyper construct' does not seem to be clear. Please indicate which it is in the photo.
2. Osteoblasts expressed lower levels of osteogenic genes. Why is that? Are the osteoblasts already past the peak of osteogenic differentiation, or were there more non-osteoblasts in the graft? Please have a little discussion about that.
3. Osteo-constructs released in the conditioned media the higher levels of cytokines related to osteoclastogeneis and bone resorption. For osteoclastogenesis, RANKL is known to be a very important factor, did you not consider soluble RANKL release?
4. In Figure 4, scaffold mineralization is shown in bone samples. However, in hyper samples, scaffolds do not appear to be mineralized. Is it possible to consider that this difference allows the osteo-grafts to shape the bone better to follow the structural pattern of the scaffold? Additionally, is the scaffold eventually resorbed?
5. Line 300: “ANOVA” is already indicated on line 296.
6. Spaces are required between numbers and some mathematical symbols (=, <).
7. This is just an interest: Chondrocytes can differentiate under lower partial pressure of oxygen than osteoblasts. This difference in cell properties may be the reason for the regenerating potential of the two grafts. More oxygen supplied to the cells in the 3D silk scaffolds might have encouraged more osteoblasts to regenerate bone faster.
Round 2
Reviewer 1 Report
The author replied my questions, using citations and reasons of not adding a blank control. but it is acceptable.
Author Response
We would like to thank the Reviewer one more time for most constructive comments and this final favorable evaluation.
Reviewer 2 Report
Dear Authors,
Although I am mostly satisfied with your answers and corrections, there is one question.
In your answer to the fourth comment, you wrote "We now include discussion of the scaffold degradation (please see Page 18, lines 616 -620)." I couldn't recognize which sentence it meant. I'm sorry if I just missed it.
I'd be interested to know what additional discussion you added, so please let me know.
Author Response
We apologize - the additional discussion was included, but we missed to show it in blue font. The two sentences in question are:
"Evaluation of the resorption through hard bone histology demonstrated instances of scaffold resorption caused by mononucleated cells indented at the scaffold surfaces throughout the osteo constructs. However, the lack of multinucleated giant cells, indicators of rapid degeneration, suggest a slow silk scaffold resorption and matrix turnover"
and they can be found on page 18, lines 612-616 of the final version of the manuscript.
We would like to again thank the Reviewer for their insightful and most helpful comments.